# Future Issues in Global Health: Challenges and Conundrums

**DOI:** 10.3390/ijerph22030325

**Published:** 2025-02-21

**Authors:** Manoj Sharma, Md Sohail Akhter, Sharmistha Roy, Refat Srejon

**Affiliations:** 1Department of Social and Behavioral Health, School of Public Health, University of Nevada, Las Vegas, NV 89119, USA; 2Department of Internal Medicine, Kirk Kerkorian School of Medicine at UNLV, University of Nevada, Las Vegas, NV 89102, USA

**Keywords:** global health, health inequities, non-communicable diseases (NCDs), infectious diseases, climate change and health, sustainable development goals (SDGs)

## Abstract

This perspective lays out the challenges and conundrums facing global health and discusses possible solutions applicable in the future. The world is facing numerous challenges that include those associated with globalization, climate change, emerging diseases, continuation of non-communicable diseases, reemerging communicable diseases, antimicrobial resistance (AMR), wars, terrorism, and humanitarian crises, among others. The recent challenges exaggerated by the COVID-19 pandemic have exposed vulnerabilities within healthcare systems, particularly in low- and middle-income countries (LMIC). The solutions must be interprofessional and multifarious with collaborative efforts and partnerships. One world order seems to be a far-fetched ideal utopian goal, but it can be a remedy for ensuring health for all. In the meantime, strengthening the World Health Organization’s role in coordinating global health efforts and improving its capacity to respond to future health crises will be critical in ensuring that the vision of a unified, healthier world becomes a reality.

## 1. Introduction

The current global health landscape is characterized by a complex interplay of longstanding and emerging challenges that strain healthcare systems worldwide. These challenges include the pervasive rise in non-communicable diseases (NCDs), infectious disease outbreaks, and environmental health risks. The COVID-19 pandemic has underscored the vulnerability of even well-resourced health systems, exposing weaknesses in healthcare infrastructure and amplifying existing health inequities [1]. Health disparities remain stark, with low- and middle-income countries (LMICs) disproportionately affected by the dual burden of infectious diseases and a growing prevalence of chronic health conditions. These countries often struggle with limited healthcare resources and access issues, which have been exacerbated by the economic and social disruptions caused by the pandemic [2]. Addressing this multifaceted health landscape requires collaborative global health strategies that prioritize preventive measures, strengthen health systems, and promote equitable access to care.

The United Nations Sustainable Development Goals (SDGs), adopted in 2015, provide a global framework for achieving health equity and sustainable development by 2030. Health is central to this agenda, particularly through Goal 3, which aims to ensure healthy lives and promote well-being for all ages [3]. However, progress toward the health-related SDGs has faced significant setbacks, especially following the COVID-19 pandemic, which disrupted health systems, exacerbated health inequities, and diverted resources from critical healthcare services and infrastructure [4]. These challenges have impacted not only SDG 3 but also goals related to poverty (SDG 1), hunger (SDG 2), clean water and sanitation (SDG 6), and reduced inequalities (SDG 10), as they are intrinsically linked to health outcomes and quality of life [5].

Despite progress in some areas, such as reductions in maternal and child mortality, the world remains off track in meeting many SDG health targets. For instance, non-communicable diseases (NCDs) like cardiovascular disease, hypertension, cancer, and diabetes, responsible for over 80% of global deaths, continue to rise, especially in low- and middle-income countries [6]. According to the GBD 2021 report, high blood pressure, tobacco use, dietary risks, and air pollution are among the top contributors to the global burden of disease. These modifiable risk factors drive the prevalence of NCDs, affecting millions worldwide and placing significant pressure on healthcare resources [7]. For instance, tobacco use remains a primary risk factor for cancers and respiratory diseases, while poor dietary habits contribute to cardiovascular conditions and metabolic disorders. This shift from infectious diseases to chronic health conditions signals an epidemiological transition fueled by lifestyle changes, urbanization, and aging populations [8]. NCDs have increasingly affected LMICs, where healthcare systems are less equipped to manage long-term chronic diseases. Similarly, infectious diseases, including HIV/AIDS, tuberculosis, and malaria, remain prevalent in regions with high poverty rates, while antimicrobial resistance poses a growing threat to disease control [2,9]. In addition, the intersection of climate change and health has gained attention, as environmental factors such as air pollution and extreme weather events contribute significantly to disease burden and mortality, particularly among vulnerable populations [10]. This convergence of multiple health challenges has galvanized the need for sustainable, resilient healthcare systems capable of addressing both current and future public health crises.

Achieving the SDGs requires renewed commitment from governments, international organizations, and communities to address structural health inequities and invest in strengthening health systems. Interventions such as improving access to essential healthcare services, expanding health workforce capacity, and implementing policies to tackle behavioral and environmental health risks are crucial. The GBD study’s insights into global disease burdens and risk factors provide a data-driven foundation for guiding these efforts and aligning policy initiatives with the SDG targets. Considering current challenges, the SDGs represent both an opportunity and a call to action to create a healthier, more equitable world, underscoring the need for integrated, multisectoral approaches to public health. Against this backdrop, this article analyzes the world’s current health and health-related challenges and projects future issues focusing on finding preemptive solutions for a healthy world.

Challenges of globalization, while facilitating interconnectedness, have also introduced a series of economic, political, and technological challenges that complicate sustainable development and equitable progress [10]. Economically, globalization has led to growth in international trade and wealth, yet it has also deepened inequalities, where wealthier nations often benefit more than developing ones. Job displacement and labor exploitation are common, as companies seek cheaper labor overseas, sometimes under poor working conditions. Poverty remains pervasive in many regions, especially in low- and middle-income countries, where populations are disproportionately affected by limited access to resources and social safety nets, impeding their economic growth and ability to adapt to the global economy [11].

Politically, globalization has fostered polarization within and between countries. Internally, many nations face tensions between nationalistic movements and proponents of international cooperation, resulting in deep political divides and the erosion of social cohesion [12]. Externally, the disparity between developed and developing countries often leads to unequal power dynamics and policy conflicts, particularly in areas like trade, environmental standards, and human rights [12]. This polarization hinders the ability of countries to form effective alliances and take unified actions on global issues such as climate change and migration.

Technologically, advancements in communication, artificial intelligence (AI), and stem cell research offer transformative possibilities but also pose challenges [13]. Rapid communication advances have intensified the spread of misinformation and increased data privacy concerns, while AI raises ethical issues regarding automation, surveillance, and job displacement [14]. Furthermore, stem cell research, though promising for medical advancement, is mired in ethical debates about its applications and accessibility, potentially widening healthcare inequities between wealthy and poorer nations [14].

Addressing these challenges requires concerted global efforts and sustainable solutions. Economically, countries can collaborate to create fairer trade practices and provide support for poverty reduction programs, aiming to close the wealth gap and ensure that globalization benefits all. Stronger labor laws and ethical sourcing standards can mitigate labor exploitation, improving working conditions worldwide. Politically, strengthening multilateral organizations and promoting diplomacy can help countries navigate polarization and work towards common goals, such as climate action and public health initiatives. To preserve cultural diversity, initiatives that support local arts, languages, and traditions are vital. Technological challenges call for ethical frameworks and regulatory standards to ensure that advances like AI and stem cell research are used responsibly and inclusively [14]. The United Nations and other international organizations play a critical role in coordinating these efforts, fostering collaboration, and encouraging policy coherence across borders. Strengthening cybersecurity and data protection laws internationally will also help secure the digital landscape, ensuring a safer and more inclusive approach to globalization. This will help create a foundation for a more equitable and sustainable future. In this article, we first discuss the challenges of climate change, followed by the challenges of emerging diseases, the challenges of reemerging communicable diseases, the challenges of non-communicable diseases, the challenges of natural disasters, and finally, wars, terrorism, and humanitarian crises. We discuss solutions through global partnerships.

## 2. Challenges of Climate Change

The change in global temperature and weather patterns over a period can be referred to as climate change. The attributes of climate change are no longer confined to natural phenomena. Human activities, particularly extensive use of fossil fuels, deforestation, and industrial processes, are causing drastic climate change, leading to unpredictable impacts globally. The Intergovernmental Panel on Climate Change (IPCC) demonstrated that the current rapid rate of climate change is extensively driven by human-induced factors, resulting in increased greenhouse gas emissions that trap atmospheric heat [15]. The effects of climate change not only affect the environment but also the lives on the earth. It severely impacts global health, food security, and water resources. Compounding the problem of climate change is the problem of air quality in some countries. Smog is also a rising concern for health in countries where the Air Quality Index (AQI) exceeds dangerous limits.

### 2.1. Causes and Effects of Climate Change

On top of natural climate change, unexpected anthropogenic factors are contributing to drastic climate change. The underlying anthropogenic factors include the combustion of fossil fuels that release greenhouse gases, including carbon dioxide (CO_2_) in the atmosphere; deforestation due to urbanization, and food production for population expansion that minimizes greenhouse gas absorption [15,16]. With the growth of civilization, industrialization has advanced drastically in every sector. Industrialization of agriculture, including livestock production and rice cultivation, has significantly contributed to greenhouse gas emissions, including methane [17].

The effects of climate change are diverse and multifaceted. The increased global temperature has resulted in more frequent severe weather calamities such as hurricanes, floods, and droughts, resulting in extermination of communities and food production [18]. The increase in global temperature due to climate change has reduced global food production, with a drop in food production of one-third predicted by 2050 [18]. Excessive environmental temperatures destabilize biological functions, leading to autoimmune and chronic diseases such as cancer, diabetes, respiratory problems, and vector-borne diseases [15,19].

### 2.2. Solutions and Global Actions

Addressing climate change requires multifaceted approaches, including mitigation, adaptation, and global cooperation. Mitigation strategies recommend using renewable energy, enhancing energy efficiency, and practicing sustainable agriculture to reduce greenhouse gas emissions [15,20]. Natural ecosystem restoration activities such as reforestation and wetland restoration have been identified as effective carbon sequestration methods [21]. Integrated actions are required across the world to address climate change. Global leaders are congregating annually to discuss possible solutions. The Paris Agreement in 2015 represents a landmark commitment by all the nations to limit global warming below 2 °C above the pre-industrial temperature [15]. All countries are required to develop unique action plans to address the national level of issues impacting climate change. Community engagement and action are required to implement the proposed action plan, and they can be achieved through public awareness and education from a fundamental level [17]. Collaborative efforts, including governmental and non-governmental organizations and local communities, can promote sustainable, effective practices to address climate change [22].

## 3. Challenges of Emerging Diseases

Some noteworthy emerging public health diseases include coronavirus infections, which have affected regions globally, with significant outbreaks in Asia (particularly China), Europe, North America, and the Middle East. *E. coli* O157:H7 has been a concern in North America and Europe, with various outbreaks reported in these areas. The Human Immuno-Deficiency Virus (HIV) remains a global issue, with particularly high prevalence in Sub-Saharan Africa, as well as in Asia, Latin America, and North America. Lyme disease is mostly found in North America, especially in the northeastern and midwestern United States, as well as in parts of Europe. Monkeypox has primarily been reported in Central and West Africa, but there have been recent outbreaks in North America and Europe. Hantavirus is most prevalent in North America, particularly the Western United States, as well as in South America, including Chile and Argentina, and parts of Asia. West Nile Virus, which affects regions worldwide, is notably present in the Middle East, North America, Africa, and parts of Europe. Dengue fever, a disease that thrives in tropical and subtropical regions, is widespread in Southeast Asia, Latin America, and parts of Africa. Zika virus, found primarily in tropical and subtropical regions, has caused significant outbreaks in South America, Southeast Asia, and the Pacific Islands. These diseases pose unique challenges to public health, with varying levels of impact in different parts of the world.

Early detection of infectious diseases such as COVID-19, HIV, and Lyme disease is crucial for effective public health responses. Rapid identification of cases allows for timely intervention, reducing transmission rates and healthcare burdens [23,24,25]. Health systems that are responsive and adaptable can implement effective surveillance and control measures, which are essential during outbreaks [26,27]. For instance, community-based surveillance has proven effective in detecting localized outbreaks, thereby enhancing early warning systems [28,29]. Moreover, mitigating reporting biases is vital for accurate disease monitoring and response. Underreporting can occur due to stigma, lack of awareness, or inadequate health infrastructure, leading to a false sense of security and delayed interventions [30,31]. By improving data collection methods and engaging communities in reporting, health systems can ensure a more accurate representation of disease prevalence, facilitating better resource allocation and response strategies [32,33].

The upsurge of coronaviruses, especially SARS-CoV, SARS-CoV-2 (COVID-19), and MERS-CoV, has highlighted shortcomings in international health systems. Millions of people have died, and the global economy has been severely disrupted due to the unprecedented effects of COVID-19 [34]. These viruses spread quickly because of human-to-human transmission and their zoonotic origins. It is necessary to make strong surveillance and quick response systems to prevent outbreaks [35,36].

HIV remains a serious public health challenge, especially in the regions where the prevalence of HIV is high. The complexity of HIV infections is more complicated due to the existence of multiple strains (HIV-1 and HIV-2). In addition, disease progression and mortality rates among impacted groups are exacerbated by co-infections with other pathogens [37].

Lyme disease, caused by the bacterium Borrelia Burgdorferi, has increased in daily incidence, particularly in Europe and North America. This disease is linked to climate change and affects ticks and their host distributions [38]. As early symptoms have similarities with other diseases, they cause challenges in diagnosis and treatment and delays in proper treatment. *Escherichia coli* (*E. coli*) 0157:H7 is linked to serious foodborne outbreaks and causes hemolytic uremic syndrome, particularly in vulnerable populations, including children and the elderly [38]. Preventing these outbreaks requires raising awareness and ensuring food safety.

Hantavirus pulmonary syndrome, transmitted via rodent excreta, presents a significant threat and can lead to severe respiratory difficulty and high mortality rates. Environmental change and encroachment into wildlife habitats have increased the risk of its transmission [38]. Aedes mosquitoes are the vectors of dengue fever, which is dramatically increasing its incidence globally. Increased urbanization, climate change, and travel change contribute to the disease’s resurgence [38]. West Nile Virus transmitted by mosquitoes is still a threat in North America and other parts of the world. This virus can result in serious neurological illness, particularly affecting the elderly [38]. Public health programs focusing on raising public awareness and mosquito control are essential for prevention.

The Zika virus has emerged as a major public health issue, especially after the 2015 outbreak in Brazil. It is linked to severe birth defects, such as microcephaly [39]. Due to the complexities of this virus’s spread, including sexual transmission, prevention efforts have become more difficult. Educational campaigns and vector control strategies can mitigate the spread of the Zika virus [40,41].

## 4. Challenges of Reemerging Communicable Diseases

Reemerging diseases refer to conditions that have previously declined in incidence but are now reappearing due to complex, interrelated factors, such as antibiotic resistance, environmental changes, and a shift in public health policy. Malaria, polio (e.g., Pakistan and Afghanistan), tuberculosis, cholera, pertussis, influenza, pneumococcal disease, gonorrhea, and antimicrobial resistance (AMR) are notable reemerging communicable diseases and associated issues.

The challenges posed by reemerging communicable diseases are exacerbated by global interconnectedness, which facilitates their spread across borders. Factors such as increased travel, trade, and urbanization contribute to the rapid transmission of diseases like tuberculosis and cholera, which can reemerge in populations that may have previously controlled them [42]. This interconnectedness not only aids in the dissemination of pathogens but also complicates their identification and treatment. For instance, the use of advanced technologies, such as genomic sequencing and predictive modeling, can enhance surveillance and response strategies, allowing for quicker identification of outbreaks and more effective containment measures [43]. Moreover, the integration of data from various sources, including health systems and environmental monitoring, can improve our understanding of disease dynamics and inform public health interventions [44]. By employing innovative approaches, such as community-based surveillance and digital health tools, health authorities can better track the emergence of diseases and respond proactively [45].

The incidence and mortality rate of malaria continues to increase, particularly in Sub-Saharan Africa, making it a significant public health challenge. Malaria has reemerged in some regions due to factors including climate change, drug resistance, and disruptions in healthcare systems despite advancements in treatment and prevention [46]. Another critical reemerging disease is tuberculosis (TB), exacerbated by the increase in multidrug-resistant strains. According to the WHO, globally, TB remains one of the top fatal diseases, with an estimated 1.5 million deaths per year [47,48].

Cholera caused by Vibrio Cholerae has reemerged globally, especially in natural disasters and conflict-affected regions, which are transmitted through contaminated food and water, presenting challenges in areas where sanitation and water quality are poor. Pertussis incidence is resurging in countries with low vaccination rates. It is difficult to prevent due to new strains of bacteria and declining immunity. Seasonal influenza epidemics continue to pose a significant threat, with substantial morbidity and mortality because of novel influenza strains (H1N1 and H3N2), which present difficulties in vaccine development [49].

Antimicrobial resistance (AMR) and serotype emergence, which are not covered by existing immunization, lead to the reemergence of pneumococcal illness caused by Streptococcus pneumoniae, which is more common in children and the elderly. Management must address antibiotic resistance and develop new vaccinations [50]. Gonorrhea has reemerged due to its complicated treatment options and the risk of multi-drug-resistant strains of Neisseria gonorrhoeae. To combat this resurgence, comprehensive sexual educational programs, routine screening, and new treatment strategies are essential [51].

Antimicrobial resistance (AMR), an increasing global health concern, impacts the treatment of many infectious diseases. This poses a challenge in managing NCDs, mainly when infections complicate chronic NCD conditions. For patients with NCDs, overuse and misuse of antibiotics introduce resistant bacterial strains that can complicate treatment approaches and elevate morbidity [52]. A coordinated global response, including public education, funding for new research about new antibiotics, alternative treatment, and stewardship programs, are urgently needed to combat this problem [53].

## 5. Challenges of Non-Communicable Diseases

As a major cause of mortality and morbidity worldwide, non-communicable diseases (NCDs) account for a significant global health challenge. NCDs are illnesses that cannot spread from one person to another, including long-term problems such as heart disease, diabetes, cancer, chronic respiratory diseases, etc. Numerous underlying factors, such as lifestyle, socioeconomic conditions, and environmental impacts, are responsible for increasing the rate of NCDs.

According to the World Health Organization [16], NCDs are responsible for more than 38 million deaths annually, with a significant percentage of those fatalities taking place in LMICs. The WHO has emphasized the urgent need for effective strategies to address them [17]. Lifestyle variables such as alcohol consumption, physical inactivity, smoking, and lack of good habits are closely associated with the increased incidence rate of NCDs [54]. According to the World Health Organization (WHO), NCDs are responsible for approximately 71% of all deaths globally, with cardiovascular diseases, cancers, respiratory diseases, and diabetes being the most prevalent [55]. In Sub-Saharan Africa, the burden of NCDs has increased dramatically, with studies indicating that these diseases accounted for 43% of the total disease burden in the region by 2017 [55].

The burden of NCDs varies significantly, with low-income groups being disproportionately impacted. Higher rates of mortality and morbidity from NCDs result from these communities often facing difficulties in accessing healthcare services [56]. The double burden of disease is particularly common in LMICs, where people have both communicable and non-communicable diseases [57]. Addressing these health disparities is urgent for efficient NCD management.

The increase in NCDs has been connected to environmental changes, such as pollution, urbanization, and climate change. Exposure to poor air quality and environmental toxins can exacerbate cardiovascular conditions and respiratory diseases [58]. In addition, the development of NCDs is influenced by the lifestyle changes associated with living in urban areas, such as a rise in processed food intake and lack of physical activity. Moreover, NCDs are also associated with indoor air quality in housing where people spend most of their time.

Research indicates that educational programs focusing on lifestyle modifications can significantly improve knowledge and practices among populations, particularly adolescents, thereby preventing lifestyle disorders [59]. However, the successful implementation of these programs faces several challenges. For instance, many existing health education systems do not adequately emphasize practical applications of health-related knowledge, which can hinder behavior change [60].

## 6. Natural Disasters

Drastic climate change has resulted in frequent natural calamities that cause disasters. This havoc is not only limited to the destruction of property but also includes health emergencies such as the emergence and reemergence of infectious diseases and scanty drinking water. Human-induced climate change has increased the frequency and severity of natural disasters, which makes it essential for a robust preparedness plan coordinating with health systems, environmental management, and community engagement.

The economic losses from these disasters have been escalating, with the World Bank estimating annual damage from natural disasters at approximately USD 520 billion [61]. Floods are particularly notable for their frequency and economic impact. They account for about 91% of all documented disasters caused by natural hazards, leading to significant property damage, disruption of economic activities, and loss of life [62]. These disasters often disrupt sanitation systems, displace populations, and damage healthcare infrastructure, creating conditions conducive to disease outbreaks.

Natural disasters like floods, earthquakes, tsunamis, and cyclones may spread infectious diseases. Natural disasters and floods may contaminate water, leading to a risk of waterborne diseases such as cholera and leptospirosis. Disaster-associated shelter houses are overcrowded places that usually lack proper sanitation services and healthcare systems; these inefficiencies together facilitate the transmission of communicable diseases [15,18]. Therefore, a critical understanding of disease dynamics followed by disaster is essential to effectively prepare for disaster management.

### Strategies for Disaster Preparedness

Among the strategies for disaster preparedness, the following may be prioritized:

Integrated health systems: Understanding the aftermath of natural disasters like the 2004 Indian Ocean tsunami implies the essentiality of coordinated health systems. Integrated health systems refer to the coordination and strengthening of surveillance systems for quick disaster response, with well-trained healthcare workers ensuring the highest level of healthcare facilities, irrespective of patient loads [18].

Community Engagement and Education: Community engagement and education are crucial for disaster preparedness because a well-informed community can take proactive steps and follow the disaster effectively [16,56]. Multi-Hazard Preparedness Plans: The development of multi-hazard preparedness plans for disasters like the COVID-19 pandemic, which requires the intersection of both health emergencies and natural disasters simultaneously, is essential [16,35]. Research and Surveillance: To develop proactive approaches, continuous research to establish the relationships among environmental changes, disaster events, and health emergencies by implementing a robust surveillance system is essential [17]. Policy and Governance: Governments should make policies integrating health consideration and environmental and disaster management into disaster risk reduction strategies for disaster preparedness [20].

## 7. Wars, Terrorism, and Humanitarian Crises

Wars, terrorism, global crises, and humanitarian emergencies are critical issues that have significant impacts on global health, security, and development. These complex challenges require comprehensive understanding and multifaceted approaches to address their underlying causes and mitigate their devastating consequences. Recent data highlight the alarming scale and severity of these issues. According to a report by Human Rights Watch, there were close to 1800 incidents of organized violence reported between 2012 and 2013 alone, particularly prevalent in regions such as the Middle East, Africa, and Asia, with countries like Syria, Afghanistan, Iraq, the Democratic Republic of Congo, South Sudan, and the Central African Republic experiencing some of the most violent conflicts in recent times [63,64]. The impact of these conflicts on public health and healthcare systems is profound. The World Health Organization (WHO) has emphasized that migrants often face barriers to accessing healthcare services, which can exacerbate existing health conditions and hinder disease prevention efforts [65]. Conflicts can disrupt immunization programs, leading to the re-emergence of vaccine-preventable diseases like polio [66]. They can also result in the destruction of healthcare infrastructure, disruption of essential services, and the displacement of populations, all of which contribute to increased mortality and morbidity [67,68]. Furthermore, conflicts can exacerbate existing health disparities and create new ones, with vulnerable populations, such as women and children, bearing the brunt of the consequences [67,69]. Beyond the immediate humanitarian crisis, wars, terrorism, and global crises also have far-reaching economic and social consequences. They can lead to the destruction of infrastructure, disruption of trade and investment, and the displacement of populations, all of which can have long-lasting impacts on economic development and social stability [70,71]. Terrorism has been shown to erode social trust and cohesion, further exacerbating societal divisions and instability [72]. Addressing these complex challenges requires a multifaceted approach that combines conflict resolution, peacebuilding, and humanitarian assistance with long-term strategies for sustainable development, climate change adaptation, and global cooperation [73,74,75,76]. Improving the quality and accessibility of healthcare services in conflict-affected regions is also crucial, as it can help mitigate the immediate and long-term health consequences of these crises [67,68,77].

The ongoing war in Ukraine, which began with the Russian invasion in February 2022, has had significant public health consequences. The conflict has disrupted the country’s healthcare system, leading to shortages of medical supplies, the destruction of healthcare facilities, and the displacement of healthcare workers [78,79,80]. This has made it increasingly difficult for Ukrainians to access essential medical services, including routine vaccinations and treatment for chronic conditions [81]. The war has also had a profound impact on the mental health of the Ukrainian population, particularly among children and adolescents. Studies have shown that exposure to war can lead to an increased risk of post-traumatic stress disorder (PTSD), depression, and other mental health issues [82]. To address this, the Ukrainian government and international organizations have worked to establish mental health support services, including hotlines and counseling programs [83].

The conflict between Israel and Hamas in the Gaza Strip has also had significant public health implications. The repeated cycles of violence have destroyed healthcare infrastructure, disrupted essential services, and displaced populations [55]. This has made it challenging for residents of Gaza to access basic medical care, including treatment for chronic conditions and emergency services. The psychological impact of the conflict has also been significant, with studies showing high rates of PTSD, anxiety, and depression among the civilian population, particularly children [84,85,86]. To address these issues, healthcare providers and humanitarian organizations have worked to establish mental health support services and provide trauma-informed care to those affected by the conflict.

Terrorism, including bioterrorism and biowarfare, as well as global crises, pose significant threats to public health and healthcare systems worldwide. These complex challenges require a comprehensive understanding and multifaceted approaches to address their underlying causes and mitigate their devastating consequences. Recent data highlight the alarming scale and severity of these issues. According to a report, there were close to 1800 incidents of organized violence reported between 2012 and 2013 alone, particularly prevalent in regions such as the Middle East, Africa, and Asia [87]. Bioterrorism has become an increasing concern, with the potential for the deliberate release of dangerous pathogens causing widespread illness and death [88,89,90]. The impact of these threats on public health and healthcare systems is profound. Terrorist attacks can disrupt essential services, including immunization programs, leading to the re-emergence of vaccine-preventable diseases [91]. They can also destroy healthcare infrastructure and the displacement of populations, all of which contribute to increased mortality and morbidity [92,93]. Furthermore, the psychological impact of terrorism and global crises can be significant, with studies showing high rates of post-traumatic stress disorder, anxiety, and depression among affected populations [92]. To address these challenges, public health authorities must work closely with law enforcement, emergency response, and intelligence agencies to implement comprehensive strategies for early detection, rapid response, and effective mitigation [88,90,93]. This includes strengthening disease surveillance, improving diagnostic capabilities, and enhancing the resilience of healthcare systems [88,90,93]. Additionally, addressing the underlying social, economic, and political factors that contribute to the rise of terrorism and global crises is crucial [87,94]. This may involve addressing issues such as poverty, inequality, and social marginalization, as well as promoting conflict resolution, peacebuilding, and sustainable development [87].

The ongoing humanitarian crisis in the Gaza Strip is a pressing public health concern that requires urgent attention and coordinated global action. The region has been subjected to a prolonged siege and repeated military conflicts, leading to the destruction of healthcare infrastructure, disruption of essential services, and the displacement of populations [95,96,97]. The impact on public health has been devastating. The overcrowded and resource-constrained environment in Gaza has made the population highly vulnerable to the spread of infectious diseases, such as the recent COVID-19 pandemic [98]. The lack of access to clean water, sanitation, and essential medical supplies has also contributed to the prevalence of preventable illnesses, although specific outbreaks of cholera have not been reported in Gaza in recent years [99]. Moreover, the psychological toll of the conflict on the civilian population, particularly children and women, has been immense. Studies have shown high rates of post-traumatic stress disorder, anxiety, and depression among Gazans [100,101,102]. The trauma and humiliation experienced by families have also had a significant impact on the social fabric of the community [102]. To address these complex challenges, a comprehensive, multifaceted approach is required. This should include the following: Immediate humanitarian aid and relief efforts to address the most pressing needs, such as the provision of essential medical supplies, clean water, and sanitation services [95,96,97].Investments in the reconstruction and rehabilitation of the healthcare system, including the rebuilding of damaged facilities and the training of healthcare workers [96,97].Psychological support and trauma-informed care for the affected population, particularly vulnerable groups such as children and women [100,101,102].Advocacy for the protection of human rights and the implementation of international humanitarian law to ensure the safety and well-being of the civilian population [95,97,103].Diplomatic efforts to find a sustainable political solution to the conflict, addressing the underlying causes of the humanitarian crisis [95,97,104]. The international community has a moral and ethical obligation to respond to the humanitarian emergency in Gaza. By working collaboratively and prioritizing the health and well-being of the affected population, we can work towards a more just, equitable, and peaceful future for all.

## 8. Global Partnerships as a Solution

Global partnerships and strengthening the United Nations (UN) are crucial in addressing the complex challenges facing the world today, including wars, terrorism, global crises, and humanitarian emergencies. The UN, as the primary intergovernmental organization, has a critical role to play in fostering international cooperation and coordinating global efforts to tackle these issues [105,106]. The UN’s Sustainable Development Goals (SDGs) provide a comprehensive framework for addressing the social, environmental, economic, and political determinants of health [107,108]. By strengthening global partnerships and collaboration, the international community can work towards achieving these goals and improving the overall well-being of populations worldwide [105,108,109]. Effective global health governance requires the involvement of a diverse range of stakeholders, including national governments, international organizations, civil society, and the private sector [106,110]. The UN can serve as a platform for facilitating these partnerships and ensuring that the voices of all stakeholders are heard [106,110]. Moreover, the UN can play a crucial role in strengthening national public health systems and capacities, particularly in conflict-affected and resource-constrained regions [105,111]. This includes supporting the development of national public health laws, improving disease surveillance and response mechanisms, and enhancing the resilience of healthcare systems [105,111]. Nurses and other healthcare professionals can also contribute to the UN’s efforts by engaging in global health diplomacy, advocating for the protection of human rights, and participating in the implementation of the SDGs [112,113]. By leveraging their expertise and amplifying their voices, healthcare professionals can help shape global health policies and ensure that the needs of vulnerable populations are addressed [112,113].

The World Health Organization (WHO) plays a crucial role in global health governance, serving as the leading intergovernmental organization responsible for coordinating international efforts to address public health challenges [114]. However, the COVID-19 pandemic has exposed the fragility of the current global health system, the need for strengthening multilateral partnerships, and the WHO’s capacity to respond effectively to emerging health crises [114,115]. One key area for improvement is enhancing the WHO’s ability to coordinate and mobilize resources during public health emergencies. The organization’s response to the Ebola outbreak in West Africa in 2014–2016 was widely criticized for its slow and ineffective coordination, leading to calls for reforms to improve its emergency response capabilities [115]. The COVID-19 pandemic has further highlighted the need for the WHO to have greater authority, funding, and access to real-time data to enable a more rapid and coordinated global response [115]. Strengthening multilateral partnerships and collaboration between the WHO and other international organizations, national governments, civil society, and the private sector is also crucial [114]. This can involve initiatives such as the establishment of global health security frameworks, the development of shared early warning systems, and the facilitation of equitable access to essential medical supplies and technologies [114]. Moreover, the WHO can play a pivotal role in promoting universal health coverage and addressing the social determinants of health, which are critical for achieving the United Nations’ Sustainable Development Goals [86]. This may involve advocating for increased investment in primary healthcare, strengthening health systems, and addressing issues such as poverty, inequality, and climate change [86]. To achieve these goals, the WHO must also enhance its own governance and accountability mechanisms, ensuring transparency, inclusivity, and responsiveness to the needs of its member states and the global community [87]. This may involve reforms to its decision-making processes, the diversification of its funding sources, and the strengthening of its partnerships with civil society and other stakeholders [87].

The World Health Organization (WHO) plays a central role in global health governance, particularly in the management of public health emergencies and disaster response. However, the organization’s current capacity suggests a need for substantial improvements in multiple areas to effectively address the complex challenges associated with global health crises, including natural disasters. A primary concern is the WHO’s ability to coordinate and implement integrated, comprehensive disaster risk management strategies. To address this, the WHO has developed frameworks such as the Health Emergency and Disaster Risk Management (Health-EDRM) paradigm, designed to incorporate health considerations into broader disaster risk reduction initiatives. This framework aims to enhance the organization’s ability to integrate health system strengthening with disaster preparedness and response efforts, although challenges remain in its full operationalization and integration across sectors. [116]. However, the field of Health-EDRM remains fragmented and underdeveloped, with challenges such as overlapping research, lack of strategic agendas, and limited stakeholder coordination [116]. This fragmentation can hinder the WHO’s ability to respond effectively to public health emergencies, as seen during the COVID-19 pandemic, where coordination among various global health entities was critical yet often lacking [117]. Moreover, the WHO’s reliance on member states for funding and political support can limit its operational effectiveness. Many countries prioritize national interests over global health commitments, which can result in inadequate funding for WHO initiatives. This situation is exacerbated by the increasing frequency and severity of natural disasters, which require substantial resources for preparedness and response. For instance, the economic impact of disasters is significant, with estimates indicating that natural disasters caused approximately USD 1.5 trillion in damages globally between 2006 and 2015 [118]. The WHO must enhance its capacity to mobilize resources and support countries in building resilient health systems that can withstand such shocks. Furthermore, while the WHO is a leading global health agency, it is not the only organization involved in disaster response. Other entities, such as the United Nations Office for Disaster Risk Reduction (UNDRR) and various non-governmental organizations, also play crucial roles. The WHO must collaborate effectively with these organizations to leverage their expertise and resources, ensuring a comprehensive approach to disaster risk management [119].

The World Health Organization (WHO) holds a central position in global health governance, particularly in addressing public health emergencies and coordinating disaster response efforts. However, the organization’s current status reveals a need for significant improvements in various areas to effectively manage the multifaceted challenges presented by global health crises, including natural disasters. A key concern is the WHO’s ability to coordinate and implement robust disaster risk management strategies. While the WHO has developed frameworks such as the Health Emergency and Disaster Risk Management (Health-EDRM) paradigm, which aims to integrate health considerations into disaster risk reduction initiatives, the field of Health-EDRM remains fragmented and underdeveloped. This fragmentation is characterized by issues such as redundant research, absence of cohesive strategic agendas, and limited coordination among stakeholders. These challenges undermine the WHO’s ability to respond effectively to public health emergencies, as was evident during the COVID-19 pandemic, when critical coordination among global health entities was often lacking.

Additionally, the WHO’s reliance on member states for financial contributions and political backing poses limitations on its operational effectiveness. Many nations prioritize national interests over global health responsibilities, leading to insufficient funding for WHO-led initiatives. The latest declarations by the Trump Administration in the US seem to compound the problems of funding of the World Health Organization, but it is hoped that better sense will prevail, and these measures will not be implemented. This issue is compounded by the increasing frequency and intensity of natural disasters, which demand significant resources for both preparedness and response. The economic toll of such disasters is substantial, with an estimated USD 1.5 trillion in damage globally between 2006 and 2015. The WHO must strengthen its capacity to mobilize resources and assist countries in developing resilient health systems capable of withstanding these shocks. Furthermore, while the WHO is a leading global health organization, it is not the sole entity involved in disaster response. Other organizations, such as the United Nations Office for Disaster Risk Reduction (UNDRR) and numerous non-governmental organizations, play critical roles in disaster response. The WHO must foster effective collaboration with these entities to maximize their expertise and resources, ensuring a unified and comprehensive approach to disaster risk management. We summarize all the things in Table 1.

## 9. Conclusions: Call for Action and Progress Toward One World Order

The current global health landscape, shaped by a convergence of long-standing and emerging health crises, calls for an urgent call to action. The recent challenges underscored by the COVID-19 pandemic have exposed vulnerabilities within healthcare systems, particularly in low- and middle-income countries, which face the dual burden of infectious diseases and the growing prevalence of non-communicable diseases. However, these challenges also present an unprecedented opportunity for reflection. It is crucial to rethink our collective approach to global health governance, focusing on fostering sustainable, resilient healthcare systems that can withstand future shocks. Governments, international organizations, and healthcare stakeholders must shift from reactive to proactive strategies, addressing the root causes of health inequities and strengthening the capacity for rapid response. 

Reflecting on the implications of globalization, climate change, and the increasing threat of emerging infectious diseases, there is a pressing need for integrated, multisectoral approaches. The creation of equitable and universal healthcare systems should remain a top priority. This will require investing in primary healthcare infrastructure, prioritizing access to essential services, and addressing the behavioral and environmental factors driving health disparities. Moreover, greater cooperation between nations, bolstered by the United Nations’ SDGs framework, is essential in promoting policies that unite efforts toward achieving health equity and sustainable development. By harmonizing efforts and ensuring the equitable distribution of resources, the global community can make significant strides toward health for all.

The concept of a “One World Order” requires not only political and economic cooperation but also an ethical commitment to address the interconnected challenges of global health, inequality, and environmental sustainability. Continuing progress toward this utopian ideal will depend on the collaborative efforts of the international community in managing crises, such as ongoing humanitarian emergencies, and preventing future disasters. Future recommendations include the establishment of global health security frameworks, enhanced disease surveillance systems, and sustainable climate change mitigation strategies. Strengthening the World Health Organization’s role in coordinating these efforts and improving its capacity to respond to future health crises will also be critical in ensuring that the vision of a unified, healthier world becomes a reality.

## Figures and Tables

**Table 1 ijerph-22-00325-t001:** Global health challenges and strategic solutions.

Section	Key Topics	Challenges	Solutions/Global Actions
Challenges of climate change	Causes and effects of climate change, air quality, smog	Fossil fuel use, deforestation, extreme weather, food insecurity, health risks (e.g., respiratory diseases, cancer)	Renewable energy, sustainable agriculture, reforestation, Paris Agreement, public awareness, community engagement
Emerging diseases	COVID-19, HIV, Lyme disease, *E. coli*, Hantavirus, Dengue, Zika, West Nile	Rapid spread, zoonotic origins, diagnostic challenges, vaccine development, vector control	Early detection, surveillance, community-based reporting, public awareness, vector control strategies
Reemerging diseases	Malaria, TB, cholera, pertussis, influenza, AMR, gonorrhea	Drug resistance, low vaccination rates, disrupted healthcare systems, environmental changes	Improved surveillance, genomic sequencing, vaccination programs, antibiotic stewardship, public health policies
Non-communicable diseases	Cardiovascular diseases, diabetes, cancer, chronic respiratory diseases	Lifestyle factors (smoking, poor diet, inactivity), environmental pollution, healthcare access disparities	Health education, lifestyle modification programs, equitable healthcare access, pollution control
Natural disasters	Floods, earthquakes, tsunamis, cyclones	Waterborne diseases, overcrowded shelters, disrupted healthcare systems, economic losses	Integrated health systems, community education, multi-hazard preparedness plans, research, policy integration
Wars, terrorism, crises	Ukraine war, Gaza conflict, bioterrorism, humanitarian crises	Healthcare system destruction, mental health issues, displacement, vaccine-preventable disease resurgence	Humanitarian aid, mental health support, healthcare system rebuilding, conflict resolution, global cooperation
Global partnerships	UN, WHO, SDGs, global health governance	Fragmented coordination, funding shortages, political prioritization of national interests over global health	Strengthening WHO capacity, equitable resource distribution, global health security frameworks, universal health coverage

## Data Availability

No new data were created or analyzed in this study. Data sharing is not applicable to this article.

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
