# Peer review of "Future Issues in Global Health: Challenges and Conundrums"

_ijerph, 2025, doi:10.3390/ijerph22030325_

Round 1

Reviewer 1 Report

Comments and Suggestions for Authors

This paper focuses on the challenges and difficulties of some global health issues, trying to characterize the current situation to frame the solutions.

However, the question you are answering is not completely clear, as should be stated in the last paragraph of the introduction.

The listed topics are not quite new neither the description you made of them.

You make broad claims without providing specific examples or evidence to substantiate them, weakening the argument's persuasiveness. Also, it lacks robust data analysis or empirical evidence to support its conclusions.

You are redundant in several parts like the role of global partnerships and the impact of climate change, which are repeated across sections. It would benefit from a reorganization to make it more concise.

Consequently, the recommendations for addressing challenges are broad and lack actionable specificity, making it difficult to understand their practicability and translate them into real actions.

A table to resume the main topics would be great to help understand your rationale.

Comments on the Quality of English Language

The paper contains some inconsistencies in writing style and formatting, such as uneven citation practices and occasional grammatical errors.

Author Response

Comment 1: This paper focuses on the challenges and difficulties of some global health issues, trying to characterize the current situation to frame the solutions. However, the question you are answering is not completely clear, as should be stated in the last paragraph of the introduction.

Response 1: Thank you for your excellent suggestion. We have added this to the Introduction (Please see Lines 122-126):

“In this article we first discuss challenges of climate change, then the challenges of emerging diseases, next the challenges of reemerging communicable diseases, followed by challenges of non-communicable diseases, the challenges of natural disasters and finally wars, terrorism and humanitarian crises. We discuss the solutions through global partnerships.”

Comment 2: The listed topics are not quite new, neither the description you made of them.

Response 2: Thank you for your feedback. This manuscript is intended as a perspective article, aiming to synthesize existing knowledge and provide insights into current health challenges. While the topics may not be new, the goal is to offer a fresh viewpoint on sustainable health promotion.

Comment 3: You make broad claims without providing specific examples or evidence to substantiate them, weakening the argument's persuasiveness. Also, it lacks robust data analysis or empirical evidence to support its conclusions.

Response 3: Thank you for your valuable feedback. This manuscript is not based on empirical research, and as such, we did not collect primary data for analysis. Instead, the focus of this article is on synthesizing existing literature and presenting a perspective on sustainable health promotion. We appreciate your understanding of this distinction and hope the revised version meets your expectations.

Comment 4: You are redundant in several parts like the role of global partnerships and the impact of climate change, which are repeated across sections. It would benefit from a reorganization to make it more concise.

Response 4: Thank you for pointing this out. We have revised the manuscript to eliminate redundancy and reorganized the sections for better flow and clarity. Additionally, we have expanded the content to provide more depth on the role of global partnerships and the impact of climate change.

Comment 5: Consequently, the recommendations for addressing challenges are broad and lack actionable specificity, making it difficult to understand their practicability and translate them into real actions.

Response 5: Thank you for your feedback. We have revised the manuscript to make the recommendations more specific and actionable. Additionally, we have expanded into this section to provide clearer guidance on how to implement these solutions in practice.

Comment 6: A table to resume the main topics would be great to help understand your rationale.

Response 6: We created a table for manuscripts. (Please see Line 583)

Section

Key Topics

Challenges

Solutions/Global Actions

Challenges of Climate Change

Causes and effects of climate change, air quality, smog

Fossil fuel use, deforestation, extreme weather, food insecurity, health risks (e.g., respiratory diseases, cancer)

Renewable energy, sustainable agriculture, reforestation, Paris Agreement, public awareness, community engagement

Emerging Diseases

COVID-19, HIV, Lyme disease, E. coli, Hantavirus, Dengue, Zika, West Nile

Rapid spread, zoonotic origins, diagnostic challenges, vaccine development, vector control

Early detection, surveillance, community-based reporting, public awareness, vector control strategies

Reemerging Diseases

Malaria, TB, cholera, pertussis, influenza, AMR, gonorrhea

Drug resistance, low vaccination rates, disrupted healthcare systems, environmental changes

Improved surveillance, genomic sequencing, vaccination programs, antibiotic stewardship, public health policies

Non-Communicable Diseases

Cardiovascular diseases, diabetes, cancer, chronic respiratory diseases

Lifestyle factors (smoking, poor diet, inactivity), environmental pollution, healthcare access disparities

Health education, lifestyle modification programs, equitable healthcare access, pollution control

Natural Disasters

Floods, earthquakes, tsunamis, cyclones

Waterborne diseases, overcrowded shelters, disrupted healthcare systems, economic losses

Integrated health systems, community education, multi-hazard preparedness plans, research, policy integration

Wars, Terrorism, Crises

Ukraine war, Gaza conflict, bioterrorism, humanitarian crises

Healthcare system destruction, mental health issues, displacement, vaccine-preventable disease resurgence

Humanitarian aid, mental health support, healthcare system rebuilding, conflict resolution, global cooperation

Global Partnerships

UN, WHO, SDGs, global health governance

Fragmented coordination, funding shortages, political prioritization of national interests over global health

Strengthening WHO capacity, equitable resource distribution, global health security frameworks, universal health coverage

Thank you so much for your astute comments which have strengthened the manuscript.

Reviewer 2 Report

Comments and Suggestions for Authors

The manuscript provides an overview of the current state of affairs regarding global health, but it is difficult to envision any global health (or global studies, international studies) scholars, who are remotely aware of the current world news, would learn something new from the information presented in this essay. The information presented here is accurate and comprehensive, but the readership of the journal is already acutely aware of all of these challenges global health scholars and practitioners face. If this piece is meant to provide a "preface" of an edited volume or a special issue of a journal, I can see the value of the "bird's eye view" survey of the current global health-related concerns, but otherwise, I am hard-pressed to think how this essay would be helpful as a free-standing scholarly article that is meant to pose a distinct research question and answer it through collecting and analyzing the data. 

Author Response

Comment 1: The manuscript provides an overview of the current state of affairs regarding global health, but it is difficult to envision any global health (or global studies, international studies) scholars, who are remotely aware of the current world news, would learn something new from the information presented in this essay. The information presented here is accurate and comprehensive, but the readership of the journal is already acutely aware of all of these challenges global health scholars and practitioners face. If this piece is meant to provide a "preface" of an edited volume or a special issue of a journal, I can see the value of the "bird's eye view" survey of the current global health-related concerns, but otherwise, I am hard-pressed to think how this essay would be helpful as a free-standing scholarly article that is meant to pose a distinct research question and answer it through collecting and analyzing the data. 

Response 1: Thank you for your valuable suggestions. This manuscript has been submitted as a Perspective/Opinion article under the category identified by the editor for this issue on “Perspectives in Global Health.”  Hence, in this manuscript, we have elaborated the current global health topics for this special issue. Since this manuscript is not an empirical article and does not involve any primary data collection, we were unable to perform any data analysis. We appreciate your understanding and hope the revised version meets the expectations of the journal and the Special issue.

Thank you so very much for your astute comments which have strengthened the manuscript.

Reviewer 3 Report

Comments and Suggestions for Authors

I take the time to acknowledge the efforts of the authors as commendable for their perspectives on laying the groundwork for the further specific studies on an area that is quite timely.

Please see specific comments on the attached.

Author Response

Comment 1: The authors have not explained why the status of WHO shows that it needs more strengthening nor explained why WHO would be the one to tackle such a humongous task of this magnitude when there are other equally great global agencies. This needs to be expounded on.

Response 1: Thanks for your excellent suggestion. We have elaborated on this.  Please see Lines 511-555:

“The World Health Organization (WHO) plays a central role in global health governance, particularly in the management of public health emergencies and disaster response. However, the organization’s current capacity suggests a need for substantial improvements in multiple areas to effectively address the complex challenges associated with global health crises, including natural disasters. A primary concern is WHO’s ability to coordinate and implement integrated, comprehensive disaster risk management strategies. To address this, the WHO has developed frameworks such as the Health Emergency and Disaster Risk Management (Health-EDRM) paradigm, designed to incorporate health considerations into broader disaster risk reduction initiatives. This framework aims to enhance the organization’s ability to integrate health system strengthening with disaster preparedness and response efforts, although challenges remain in its full operationalization and integration across sectors. [117] . However, the field of Health-EDRM remains fragmented and underdeveloped, with challenges such as overlapping research, lack of strategic agendas, and limited stakeholder coordination [117]. This fragmentation can hinder the WHO's ability to respond effectively to public health emergencies, as seen during the COVID-19 pandemic, where coordination among various global health entities was critical yet often lacking [118]. Moreover, the WHO's reliance on member states for funding and political support can limit its operational effectiveness. Many countries prioritize national interests over global health commitments, which can result in inadequate funding for WHO initiatives. This situation is exacerbated by the increasing frequency and severity of natural disasters, which require substantial resources for preparedness and response. For instance, the economic impact of disasters is significant, with estimates indicating that natural disasters caused approximately USD 1.5 trillion in damages globally between 2006 and 2015 [119]. The WHO must enhance its capacity to mobilize resources and support countries in building resilient health systems that can withstand such shocks. Furthermore, while the WHO is a leading global health agency, it is not the only organization involved in disaster response. Other entities, such as the United Nations Office for Disaster Risk Reduction (UNDRR) and various non-governmental organizations, also play crucial roles. The WHO must collaborate effectively with these organizations to leverage their expertise and resources, ensuring a comprehensive approach to disaster risk management [120].

The World Health Organization (WHO) holds a central position in global health governance, particularly in addressing public health emergencies and coordinating disaster response efforts. However, the organization’s current status reveals a need for significant improvements in various areas to effectively manage the multifaceted challenges presented by global health crises, including natural disasters. A key concern is the WHO’s ability to coordinate and implement robust disaster risk management strategies. While the WHO has developed frameworks such as the Health Emergency and Disaster Risk Management (Health-EDRM) paradigm, which aims to integrate health considerations into disaster risk reduction initiatives, the field of Health-EDRM remains fragmented and underdeveloped. This fragmentation is characterized by issues such as redundant research, absence of cohesive strategic agendas, and limited coordination among stakeholders. These challenges undermine the WHO’s ability to respond effectively to public health emergencies, as was evident during the COVID-19 pandemic when critical coordination among global health entities was often lacking.”

Thank you so very much for your astute comments which have strengthened the manuscript.

Reviewer 4 Report

Comments and Suggestions for Authors

The draft contains a thorough overview of today's challenges in global health and the unpredictability of processes. With nearly 100 references, this essay-type study presents the most important emerging and reemerging risks, infectious and non-infectious diseases, humanitarian disasters associated with climate change and the escalation of armed conflicts. It correctly notes that the obligations undertaken 10 years ago in the target system of the Sustainable Development Agenda are hardly fulfilled due to the interconnected crises of recent years.

As a solution, the paper recommends the re-strengthening the global partnership and respecting the WHO's constitutional directing and coordinating role, which would undoubtedly be important. Unfortunately, things are not moving on this path and thus the otherwise important communication remains a bit one-sided.

It deserves analysis, for example, that multilateralism in health has been in a period of transition since COVID19, and its political, economic and academic powerbase in the global north no longer holds. It is much more difficult to reach consensus in a new multipolar world with constantly shifting alliances, increased lack of trust, reduced development assistance. What was held together - politically, financially and ideologically - by the US hegemon - is dissipating in a new ecosystem of power centers. It is unpredictable what the Trump administration will bring, the advance of nationalist-populist forces worldwide. So the world of global health should also be evaluated in the light of global political developments. All of this complicates WHO's financial and operational situation, hinders professionally expected agreements, such as the pandemic treaty.

Author Response

Dear Editor

Thanks for taking the time to review our manuscript entitled, “Future issues in global health: Challenges and conundrums” (Manuscript- 3417849). My co-authors and I have addressed all the comments and provided point-by-point responses for your consideration. The recent changes are highlighted in yellow in the revised manuscript.

Comment 1: The draft contains a thorough overview of today's challenges in global health and the unpredictability of processes. With nearly 100 references, this essay-type study presents the most important emerging and reemerging risks, infectious and non-infectious diseases, humanitarian disasters associated with climate change and the escalation of armed conflicts. It correctly notes that the obligations undertaken 10 years ago in the target system of the Sustainable Development Agenda are hardly fulfilled due to the interconnected crises of recent years.

As a solution, the paper recommends the re-strengthening of the global partnership and respecting the WHO's constitutional directing and coordinating role, which would undoubtedly be important. Unfortunately, things are not moving on this path and thus the otherwise important communication remains a bit one-sided.

It deserves analysis, for example, that multilateralism in health has been in a period of transition since COVID19, and its political, economic and academic powerbase in the global north no longer holds. It is much more difficult to reach consensus in a new multipolar world with constantly shifting alliances, increased lack of trust, reduced development assistance. What was held together - politically, financially and ideologically - by the US hegemon - is dissipating in a new ecosystem of power centers. It is unpredictable what the Trump administration will bring, the advance of nationalist-populist forces worldwide. So the world of global health should also be evaluated in the light of global political developments. All of this complicates WHO's financial and operational situation, hinders professionally expected agreements, such as the pandemic treaty.

Response 1: Thanks for your excellent suggestions. We have elaborated on this including expanded coverage on WHO’s potential role.  Please see Lines 511-555:

“The World Health Organization (WHO) plays a central role in global health governance, particularly in the management of public health emergencies and disaster response. However, the organization’s current capacity suggests a need for substantial improvements in multiple areas to effectively address the complex challenges associated with global health crises, including natural disasters. A primary concern is WHO’s ability to coordinate and implement integrated, comprehensive disaster risk management strategies. To address this, the WHO has developed frameworks such as the Health Emergency and Disaster Risk Management (Health-EDRM) paradigm, designed to incorporate health considerations into broader disaster risk reduction initiatives. This framework aims to enhance the organization’s ability to integrate health system strengthening with disaster preparedness and response efforts, although challenges remain in its full operationalization and integration across sectors. [117] . However, the field of Health-EDRM remains fragmented and underdeveloped, with challenges such as overlapping research, lack of strategic agendas, and limited stakeholder coordination [117]. This fragmentation can hinder the WHO's ability to respond effectively to public health emergencies, as seen during the COVID-19 pandemic, where coordination among various global health entities was critical yet often lacking [118]. Moreover, the WHO's reliance on member states for funding and political support can limit its operational effectiveness. Many countries prioritize national interests over global health commitments, which can result in inadequate funding for WHO initiatives. This situation is exacerbated by the increasing frequency and severity of natural disasters, which require substantial resources for preparedness and response. For instance, the economic impact of disasters is significant, with estimates indicating that natural disasters caused approximately USD 1.5 trillion in damages globally between 2006 and 2015 [119]. The WHO must enhance its capacity to mobilize resources and support countries in building resilient health systems that can withstand such shocks. Furthermore, while the WHO is a leading global health agency, it is not the only organization involved in disaster response. Other entities, such as the United Nations Office for Disaster Risk Reduction (UNDRR) and various non-governmental organizations, also play crucial roles. The WHO must collaborate effectively with these organizations to leverage their expertise and resources, ensuring a comprehensive approach to disaster risk management [120].

The World Health Organization (WHO) holds a central position in global health governance, particularly in addressing public health emergencies and coordinating disaster response efforts. However, the organization’s current status reveals a need for significant improvements in various areas to effectively manage the multifaceted challenges presented by global health crises, including natural disasters. A key concern is the WHO’s ability to coordinate and implement robust disaster risk management strategies. While the WHO has developed frameworks such as the Health Emergency and Disaster Risk Management (Health-EDRM) paradigm, which aims to integrate health considerations into disaster risk reduction initiatives, the field of Health-EDRM remains fragmented and underdeveloped. This fragmentation is characterized by issues such as redundant research, absence of cohesive strategic agendas, and limited coordination among stakeholders. These challenges undermine the WHO’s ability to respond effectively to public health emergencies, as was evident during the COVID-19 pandemic, when critical coordination among global health entities was often lacking.”

Please also see Lines 559-561:

The latest declarations by Trump Administration in the US seem to compound the problems of funding of the World Health Organization but it is hoped that better sense will prevail, and these measures will not be implemented

Thank you so very much for your astute comments which have strengthened the manuscript.

Reviewer 5 Report

Comments and Suggestions for Authors

Line 40 - Do you mean "health equity", with no comma?

Line 51 - What about hypertension which I think ranks slightly under diabetes in prevalence?

Line 134 - anthropogenic factors are contributing to drastic climate change

Line 144 - global temperature has contributed to more frequent severe weather calamities

Line 159 - At the time of this review I note that the US is likely to or has withdrawn from the Paris Agreement.  I suggest that you consider this and make any necessary amendments to your manuscript.

Line 168 - What about MPox emanating from the DR of Congo?

Line 190 - encroachment into wildlife habitats

Line 194 - effective vaccines complicate the management efforts

Line 215 - "...top fetal diseases..."  While there is no apparent error here you may wish to clarify that you do intend it to mean a condition that affects an unborn baby during pregnancy.  The alternative is that some readers my think you have made an error and that you mean "fatal diseases".

Line 234 - complicate chronic NCD conditions.

Line 249 - "poor habits" or "a lack of good habits"

Line 261 - NCDs are also associated with indoor air quality in housing where people spend most of their time.

Line 286 - Multi-Hazard Preparedness Plans.  Should this be in italics?

Line 289 - Research and Surveillance. Should this be in italics?

Line 292 - Policy and Governance. Should this be in italics?

Line 468 - that unite efforts toward achieving health equity

Author Response

Comment 1: Line 40 - Do you mean "health equity", with no comma?

Response 1: Thank you for pointing this out. Yes, we meant “health equity” and we have corrected it. (Please see Lines 40)

Comment 2: Line 51 - What about hypertension which I think ranks slightly under diabetes in prevalence?

Response 2: Thank you for pointing this out. We have added “hypertension” as per your suggestion. (Please see Lines 51)

Comment 3: Line 134 - anthropogenic factors contribute to drastic climate change.

Response 3: Thank you for pointing this out. We have corrected it as per your suggestion. (Please see Lines 141)

“Anthropogenic factors are contributing drastic climate change”

Comment 4: Line 144 - global temperature has contributed to more frequent severe weather calamities.

Response 4: Thank you for pointing this out. We have corrected it as per your suggestion. (Please see Lines 149)

“Global temperature has raised more frequent severe weather calamities”

Comment 5: Line 168 - What about MPox emanating from the DR of Congo?

Response 5: Thank you for pointing this out. We have included MPox. (Please see Lines 181)

“Monkeypox”

Comment 6: Line 190 - encroachment into wildlife habitats

Response 6: Thank you for pointing this out. We have corrected it as per your suggestion. (Please see Lines 225)

“Encroachment into wildlife habitats”

Comment 7: Line 194 - effective vaccines complicate the management efforts

Response 7: Thank you for pointing this out. We have corrected it as per your suggestion. (Please see Lines 229)

“Effective vaccines complicate the management efforts”

Comment 8: Line 215 - "...top fetal diseases..."  While there is no apparent error here you may wish to clarify that you do intend it to mean a condition that affects an unborn baby during pregnancy.  The alternative is that some readers my think you have made an error and that you mean "fatal diseases"

Response 8: Thank you for pointing this out. We have corrected it as per your suggestion. (Please see Lines 264)

“Fatal”

Comment 9: Line 234 - complicate chronic NCD conditions.

Response 9: Thank you for pointing this out. We have corrected it as per your suggestion. (Please see Lines 283)

“Complicate chronic NCD conditions”

Comment 10: Line 249 - "poor habits" or "a lack of good habits"

Response 10: Thank you for pointing this out. We have corrected it as per your suggestion. (Please see Lines 299)

“Lack of good habits”

Comment 11: Line 261 - NCDs are also associated with indoor air quality in housing where people spend most of their time.

Response 11: Thank you for pointing this out. We have corrected it as per your suggestion. (Please see Lines 317)

“NCDs are also associated with indoor air quality in housing where people spend most of their time”

Comment 12: Line 286 - Multi-Hazard Preparedness Plans.  Should this be in italics?

Response 12: Thank you for pointing this out. We have corrected it as per your suggestion. (Please see Lines 356)

“Multi-Hazard Preparedness Plan”

Comment 13: Line 289 - Research and Surveillance. Should this be in italics?

Response 13: Thank you for pointing this out. We have corrected it as per your suggestion. (Please see Lines 359)

“Research and Surveillance”

Comment 14: Line 292 - Policy and Governance. Should this be in italics?

Response 14: Thank you for pointing this out. We have corrected it as per your suggestion. (Please see Lines 362)

“Policy and Governance”

Comment 15: Line 468 - that unites efforts toward achieving health equity

Response 15: Thank you for pointing this out. We have corrected it as per your suggestion. (Please see Lines 606)

“That unites efforts toward achieving health equity”

Thank you so very much for your astute comments which have strengthened the manuscript.

Reviewer 6 Report

Comments and Suggestions for Authors

Line 121: The authors mentioned, “This will help create a foundation for a more equitable and sustainable future.” I suggest that a few lines at the end of the introduction about how authors have organized the manuscript in terms of content would be helpful for the reader.

Line 142: The authors mentioned the effects of climate change; I suggest to mention about smog also as a rising concern on health should be mentioned. Esp in countries where the Air Quality Index is exceeding dangerous limits.

Line 166: It is better to mention regions where these diseases emerged/are prevalent.

Line 167: The importance of early detection, responsive health systems, and mitigating reporting biases should be emphasized here.

Line 209: Polio may be added to the list considering increasing cases in some Asian countries, e.g., Afghanistan and Pakistan.

Line 209: Authors mentioned “associated issues.” It may be discussed how global interconnectedness plays a role in their spread and how the same concept can be utilized for their identification and treatment (for example, employing new technologies to identify and track, using prediction models etc.)

Line 248: As the authors mentioned, “lifestyle variables.”. Lifestyle modification programs and strategies (both development and implementation) can be discussed here as an important area for research and development.

Line 257: To emphasize the gravity of the situation, some statistics should be mentioned that show the rate of increase of NCDs over the past decade or so. With rapid population aging, this will pose a significant economic burden. 

Line 263 Heading 6. “Natural Diseases” should be placed after climate change, since it is a direct consequence of climate change issues.

Line 271: “Natural disasters and floods” authors should pin some statistics, for example, flood induced loss of lives in South Asia in recent years and communicable disease outbreaks following mass displacements after floods can be added to support the argument.

Line 306: Migration (both internal and external) should also be mentioned among the impacts which pose its own unique set of health challenges.

Line 370 – 397 This paragraph should proceed with the previous one from Line 345 to ensure continuity of similar content.

Author Response

Comment 1: Line 121: The authors mentioned, “This will help create a foundation for a more equitable and sustainable future.” I suggest that a few lines at the end of the introduction about how authors have organized the manuscript in terms of content would be helpful for the reader.

Response 1: Thank you for your excellent suggestion. We have added this to the Introduction (Please see Lines 122-126):

“In this article we first discuss challenges of climate change, then the challenges of emerging diseases, next the challenges of reemerging communicable diseases, followed by challenges of non-communicable diseases, the challenges of natural disasters and finally wars, terrorism and humanitarian crises. We discuss the solutions through global partnerships.”

Comment 2: Line 142: The authors mentioned the effects of climate change; I suggest mentioning about smog also as a rising concern on health should be mentioned. Esp in countries where the Air Quality Index is exceeding dangerous limits.

Response 2: Thank you for your excellent suggestion. We have added this.  Please see lines 136-139:

“Compounding the problem of climate change is the problem of air quality in some countries.  Smog also is a rising concern for health in countries where the Air Quality Index (AQI) is exceeding dangerous limits.” 

Comment 3: Line 166: It is better to mention regions where these diseases emerged/are prevalent.

Response 3: Thanks for your excellent suggestion. The following has been added (please see lines 174-191):

“Some noteworthy emerging public health diseases include coronavirus infections, which have affected regions globally, with significant outbreaks in Asia (particularly China), Europe, North America, and the Middle East. E. coli O157:H7 has been a concern in North America and Europe, with various outbreaks reported in these areas. The Human Immuno-Deficiency Virus (HIV) remains a global issue, with particularly high prevalence in Sub-Saharan Africa, as well as in Asia, Latin America, and North America. Lyme disease is mostly found in North America, especially in the northeastern and midwestern United States, as well as in parts of Europe. Monkeypox has primarily been reported in Central and West Africa, but there have been recent outbreaks in North America and Europe. Hantavirus is most prevalent in North America, particularly the western United States, as well as in South America, including Chile and Argentina, and parts of Asia. West Nile Virus, which affects regions worldwide, is notably present in the Middle East, North America, Africa, and parts of Europe. Dengue fever, a disease that thrives in tropical and subtropical regions, is widespread in Southeast Asia, Latin America, and parts of Africa. Zika virus, found primarily in tropical and subtropical regions, has caused significant outbreaks in South America, Southeast Asia, and the Pacific Islands. These diseases pose unique challenges to public health, with varying levels of impact in different parts of the world.”

Comment 4: Line 167: The importance of early detection, responsive health systems, and mitigating reporting biases should be emphasized here.

Response 4: Thanks for your excellent suggestion. The following has been added (Please lines 192-203):

“Early detection of infectious diseases such as COVID-19, HIV, and Lyme disease is crucial for effective public health responses. Rapid identification of cases allows for timely intervention, reducing transmission rates and healthcare burdens [23–25]. Health systems that are responsive and adaptable can implement effective surveillance and control measures, which are essential during outbreaks [26,27]. For instance, community-based surveillance has proven effective in detecting localized outbreaks, thereby enhancing early warning systems [28,29]. Moreover, mitigating reporting biases is vital for accurate disease monitoring and response. Underreporting can occur due to stigma, lack of awareness, or inadequate health infrastructure, leading to a false sense of security and delayed interventions [30,31]. By improving data collection methods and engaging communities in reporting, health systems can ensure a more accurate representation of disease prevalence, facilitating better resource allocation and response strategies [32,33].”

Comment 5: Line 209: Polio may be added to the list considering increasing cases in some Asian countries, e.g., Afghanistan and Pakistan.

Response 5: Thanks for your excellent suggestion. The following has been added (please see Lines 242-244):

“Malaria, polio (e.g., Pakistan & Afghanistan), tuberculosis, cholera, pertussis, influenza, pneumococcal disease, gonorrhea, and antimicrobial resistance (AMR) are notable reemerging communicable diseases and associated issues.”

Comment 6: Line 209: Authors mentioned “associated issues.” It may be discussed how global interconnectedness plays a role in their spread and how the same concept can be utilized for their identification and treatment (for example, employing new technologies to identify and track, using prediction models etc.)

Response 6: Thanks for your excellent suggestion. The following has been added (please see lines 246-259):

“The challenges posed by reemerging communicable diseases are exacerbated by global interconnectedness, which facilitates their spread across borders. Factors such as increased travel, trade, and urbanization contribute to the rapid transmission of diseases like tuberculosis and cholera, which can reemerge in populations that may have previously controlled them [42]. This interconnectedness not only aids in the dissemination of pathogens but also complicates their identification and treatment. For instance, the use of advanced technologies, such as genomic sequencing and predictive modeling, can enhance surveillance and response strategies, allowing for quicker identification of outbreaks and more effective containment measures [43]. Moreover, the integration of data from various sources, including health systems and environmental monitoring, can improve our understanding of disease dynamics and inform public health interventions [44]. By employing innovative approaches, such as community-based surveillance and digital health tools, health authorities can better track the emergence of diseases and respond proactively [45].”

Comment 7: Line 248: As the authors mentioned, “lifestyle variables.”. Lifestyle modification programs and strategies (both development and implementation) can be discussed here as an important area for research and development.

Response 7: Thanks for your excellent suggestion. The following has been added (please see lines 317-322):

“Research indicates that educational programs focusing on lifestyle modifications can significantly improve knowledge and practices among populations, particularly adolescents, thereby preventing lifestyle disorders [59]. However, the successful implementation of these programs faces several challenges. For instance, many existing health education systems do not adequately emphasize practical applications of health-related knowledge, which can hinder behavior change [60].”

Comment 8: Line 257: To emphasize the gravity of the situation, some statistics should be mentioned that show the rate of increase of NCDs over the past decade or so. With rapid population aging, this will pose a significant economic burden. 

Response 8: Thanks for your excellent suggestion. The following has been added (please see lines 301-305):

“According to the World Health Organization (WHO), NCDs are responsible for approximately 71% of all deaths globally, with cardiovascular diseases, cancers, respiratory diseases, and diabetes being the most prevalent [55]. In sub-Saharan Africa, the burden of NCDs has increased dramatically, with studies indicating that these diseases accounted for 43% of the total disease burden in the region by 2017 [55].”

Comment 9: Line 263 Heading 6. “Natural Diseases” should be placed after climate change, since it is a direct consequence of climate change issues.

Response 9: Thank you for your excellent suggestion but we believe it is a matter of preference.  This change will require substantial work to reorder references and content without a substantial gain on the content. We hope the reviewer will acquiesce to our preference of ordering.

Comment 10: Line 271: “Natural disasters and floods” authors should pin some statistics, for example, flood induced loss of lives in South Asia in recent years and communicable disease outbreaks following mass displacements after floods can be added to support the argument.

Response 10: Thanks for your excellent suggestion. The following has been added (please see lines 331-337):

“The economic losses from these disasters have been escalating, with the World Bank estimating annual damages from natural disasters at approximately $520 billion [61]. Floods are particularly notable for their frequency and economic impact. They account for about 91% of all documented disasters caused by natural hazards, leading to significant property damage, disruption of economic activities, and loss of life [62]. These disasters often disrupt sanitation systems, displace populations, and damage healthcare infrastructure, creating conditions conducive to disease outbreaks.”

Comment 11: Line 306: Migration (both internal and external) should also be mentioned among the impacts which pose its own unique set of health challenges.

Response 11: Thanks for your excellent suggestion. The following has been added (Please see lines 375-377):

“The World Health Organization (WHO) has emphasized that migrants often face barriers to accessing healthcare services, which can exacerbate existing health conditions and hinder disease prevention efforts [65].

Comment 12: Line 370 – 397 This paragraph should proceed with the previous one from Line 345 to ensure continuity of similar content.

Response 12: We have shifted this paragraph as per the suggestion. (Please see lines 443-470)

Thank you so very much for your astute comments which have strengthened the manuscript.
